# Effects of the *In ovo* Administration of the 6/85 *Mycoplasma gallisepticum* Vaccine on Layer Chicken Embryo Hatchability and Early Posthatch Performance [note 1]

**DOI:** 10.3390/ani13071228

**Published:** 2023-03-31

**Authors:** Abdulmohsen H. Alqhtani, Seyed Abolghasem Fatemi, Katie Elaine Collins Elliott, Scott L. Branton, Jeff Dwayne Evans, Edgar David Peebles

**Affiliations:** 1Department of Poultry Science, Mississippi State University, Starkville, MS 39762, USA; 2Department of Animal Production, College of Agriculture and Food Sciences, King Saud University, P.O. Box 2460, Riyadh 11451, Saudi Arabia; 3Poultry Research Unit, USDA-ARS, Starkville, MS 39762, USA

**Keywords:** embryo, *in ovo* injection, layer, *Mycoplasma gallisepticum*, strain 6/85

## Abstract

**Simple Summary:**

*Mycoplasma gallisepticum* (MG) has reduced egg production and caused other economic losses in the poultry sector. This study studied the potential use of *in ovo* vaccination of the 6/85 MG vaccine in layer embryos for subsequent early protection as well as live performance of pullets. The use of various dosages of live attenuated 6/85 MG vaccine in either the air cell or amnion, ranging from 1.73 to 1.73 × 10^4^ CFU was evaluated after 18 days of incubation. In the current study it was found that the *in ovo* vaccination of the high dosage of 6/85 MG (1.73 × 10^4^ CFU), when administered in the amnion (AM), resulted in detrimental impacts to the hatchling process as well as post-hatch live performance but increased three-week-old antibody titers. Additionally, as compared with the air cell (AC), the *in ovo* administration of the medium dosage of the 6/85 MG vaccine (1.73 × 10^2^) in the AM had promising results in terms of the hatchability, live performance and antibody titer of early posthatch pullets. Therefore, it is suggested that the 1.73 × 10^2^ CFU treatment injected in the AM may be an appropriate candidate for the pragmatic and commercial application of 6/85 MG vaccine because it did not result in embryo or posthatch chick mortalities while eliciting a humoral response.

**Abstract:**

*In ovo* administration as a possible alternative method of 6/85 MG vaccination was assessed. After 18 days of incubation (doi), the eggs were administered a particular dosage of a live attenuated 6/85 MG vaccine in either the air cell (AC) or amnion (AM). The treatments included non-injected eggs and eggs injected into the AC or AM with diluent alone as controls. Treatments also included eggs injected with diluent, which contained 1.73 × 10^2^, or 1.73 × 10^4^ CFU of 6/85 MG. Hatchability of viable injected eggs (HI) and residual embryonic mortality were determined at 22 doi. At hatch and at three weeks posthatch, one hatched chick per treatment replicate was bled and swabbed for the detection of 6/85 MG in the choanal cleft using PCR, serum plate agglutination (SPA), and ELISA methods. The results show that AC *in ovo* injection of 6/85 MG had no negative impacts on HI or on the live performance of pullets, but that it failed to provide adequate protection (*p* ≤ 0.0001) in hatchlings or three-week-old pullets. The 1.73 × 10^4^ 6/85 MG CFU dosage injected into the AM decreased the hatchability of injected eggs containing viable embryos (HI; *p* = 0.009) and was associated with a significant increase in late dead mortality (*p* = 0.001). Hatchling and three-week-old chick mortalities (*p* = 0.008) were significantly greater in the 1.73 × 10^4^ CFU-AM treatment group in comparison with the other treatment groups. In addition, the 1.73 and 1.73 × 10^2^ 6/85 MG-AM treatments had no negative effects on the hatching process or on posthatch growth, and the 1.73 × 10^2^ 6/85 MG-AM treatment was more effective in the protection of pullets against MG (*p* ≤ 0.0001) as compared with the low dosage and non-injected treatment groups. Further research is needed to examine the influence of the 6/85 MG *in ovo* vaccine on layer immune competence.

## 1. Introduction

According to USDA reports, a total of 84 million eggs were produced in the United States in 2018. A major challenge facing the commercial table egg industry is *Mycoplasma gallisepticum* (**MG**), which causes extensive economic losses due to decreased egg production [1]. The annual worldwide cost to the poultry industry due to MG infection is estimated to be between $118 and $150 million [1]. *Mycoplasma gallisepticum* is a pathogenic avian mycoplasma of the commercial layers [2]. It causes chronic respiratory disease [3,4,5,6,7,8] in chickens, and infectious sinusitis in turkeys [9]. It has been reported that MG infection can result in salpingitis [10,11], leading to a reduction in egg production [1,10,12] and egg quality [11,13,14,15,16]. It is well documented that MG reduces feed efficiency such that it can result in an increase in the cost of its control [1,17]. Eliciting an early immune response in posthatch chicks against subsequent field stain MG infection is important.

In the United States, commercial layers are raised in multiple-age systems and older birds can pass MG on to younger birds that have not developed strong immune systems [4,7,18]. *Mycoplasma gallisepticum* vaccines have been used commercially for the control of field-strain MG infections. Live MG vaccines can be more effective in eliminating egg production losses when they are applied to layers before field infections [19,20,21,22,23]. Birds vaccinated with the F-strain of MG (**FMG**) produce seven eggs per housed hen more than hens infected with unknown field strains of MG. In fact, when layers are vaccinated with FMG, egg production and quality as well as layer performance can be impacted [13]. It has been reported that when the 6/85-strain of MG (**6/85 MG**) is administered to hens at 10 weeks (**wk**) of age via spray, no negative impacts on egg production, egg size, and ovary function are observed [24]. Additionally, when pullets are inoculated at 10 wk of age with the ts-11 strain of MG (**ts-11 MG**), no impacts on egg production and egg size were noted [25]. In some states, commercial layer flocks are vaccinated with live FMG. This procedure was approved by the USDA in 1988 [26]. Live MG vaccinations such as ts-11MG (Merial Select, Gainesville, GA, USA) and 6/85 MG (Mycovac, Animal Health, Millsboro, DE, USA) have also been approved in the United States but do not provide protection against MG as well as the FMG vaccine [27,28].

The virulence of FMG is milder than that of field strains and has been commercially applied as a vaccine via sprayer or drinker at 8 wk of age or older [13]. The FMG vaccine administered in a dosage range between 1.3 and 1.3 × 10^6^ CFU has been successfully administered by *in ovo* injection to layer embryos at day (**d**) 18 of incubation (**doi**) [29,30]. It has been recommended that the *in ovo* injection of FMG can allow for a normal hatching process when 1.3 CFU (very low) and 1.3 × 10^2^ CFU (low) dosages are used. However, higher dosages can result in poor hatchling quality and hatchability, and higher embryonic and chick mortalities [29,30]. Moreover, a greater humoral immune response was observed in layer chicks when injected *in ovo* with 1.3 × 10^2^ CFU (low) to 1.3 × 10^6^ CFU (high) FMG dosages. Furthermore, when ts-11 MG vaccines were administered by *in ovo* injection at 18 doi, there was no negative impact on the hatching process, but no serological or proper transmission responses were observed [31].

The MG vaccine has been commercially administered in different sites, such as the eye, nares, or oral cavity [14,32,33] and can also be administered via water lines. However, it has been reported that vaccination administered by eye drop or spray leads to a faster immune response compared with other sites or methods [34]. In the United States, Embrex, Inc. manufactures the Inovoject automatic multi-egg *in ovo* injection machine [35]. *In ovo* injection has been used in broiler hatching eggs since the 1990s for Marek’s disease vaccine (MDV) [36]. *In ovo* injection has also been used for the delivery of different materials such as probiotics [37], vaccines [29,30,31,38,39,40,41,42,43], carbohydrates [44], and vitamins [45,46,47,48,49]. *In ovo* vaccination of chicken embryos against MG is relatively new but is a promising method to control MG infections in hatchlings during the early posthatch phase [29,30,41,50]. This method could minimize bird stress during vaccination and save labor and vaccine costs [42,43]. In fact, injecting MG vaccines *in ovo* may reduce potentially harmful effects when compared with regular MG vaccination practices. However, there are no published data evaluating the effects of the 6/85 MG vaccine delivered by *in ovo* injection on layer hatchability and survivability. It is well documented that the site of injection plays a crucial role in the efficacy of injected materials. However, the optimum dosages and sites of injection for maximum efficacy have not been evaluated. Nevertheless, *in ovo* administration of MDV has been shown to provide stronger protection to hatching broilers when injected *in ovo* in the amnion (AM) rather than in the air cell (AC) [42,43]. Moreover, the *in ovo* injection of L-ascorbic acid in the AC has been shown to increase hatchling quality [51] and the antioxidant capacity of 28-day-old broilers [52]. Thus, the different injected materials used between the studies could be the reason for the inconsistent results between the AC and AM injections. The objectives for this study were therefore to examine the effects of the *in ovo* vaccination of a live attenuated 6/85 MG vaccine administered at different dosages and at different sites (AC and AM) in live embryonated eggs on hatchability and pre- and posthatch livability, and to also investigate its effects on hatchling and posthatch immune responses.

## 2. Materials and Methods

### 2.1. Egg Incubation

Bird husbandry, handling, sampling, and euthanasia procedures were approved by a USDA-ARS Animal Care and Use Committee (USDA AUP 19-7). A total of 2160 fertile Hy-Line W-36 layer hatching eggs were obtained from a 25 wk-old MG-clean commercial breeder flock (Hy-Line Company, Mansfield, GA, USA). An Avida AH1-165-16 single stage incubator (Chick Master incubator Company, Medina, OH, USA) was used for both the setter (0 to 18 doi) and hatcher (18 to 22 doi) incubation phases. The incubational conditions were in accordance with the procedures described by Fatemi et al. [45,46].

### 2.2. Treatment Designation and Application

The treatments included non-injected eggs and eggs injected in the AC or AM with Poulvac Marek’s disease diluent (Zoetis, Parsippany, NJ, USA) alone as controls. Treatments also included eggs injected with diluent containing one of three different levels of 6/85 MG (NOBILIS^®^ MG6/85, Intervet International, Boxmeer, The Netherlands), which contained 1.73, 1.73 × 10^2^, or 1.73 × 10^4^ CFU of 6/85 MG. The 6/85 MG vaccine was resuspended and diluted in 50 mL of diluent. The 1 × 10^−6^ dilution of the live 6/85 MG vaccine prepared on the same day of injection contained 1.73 CFU of the organism in each 50 μL volume of solution that was injected into each egg. In addition, the 1 × 10^−4^ and 1 × 10^−2^ dilutions, respectively, contained 1.73 × 10^2^ and 1.73 × 10^4^ CFU of the organism in each 50 μL volume of solution that was injected into each egg (Table 1).

Groups of 30 eggs belonging to each of 9 pre-assigned treatment groups were randomly arranged within each of 8 replicate tray levels of the setter unit. At 12 doi, infertile eggs and those containing dead embryos were removed and only those eggs containing viable embryos were vaccinated at 18 doi [31,47]. All *in ovo* injection treatments were administered at 18 doi using an Embrex Inovoject M automated multi-egg injector (Zoetis, Parsippany, NJ, USA). Before injection, the 6/85 MG vaccine from the original vial was plated on 2 plates containing Frey’s Mycoplasma agar [53] and incubated at 37 °C to confirm vaccine viability and the actual dosage being delivered. Eggs were transferred to hatching baskets after injection. A group of 240 eggs belonging to each of 7 treatment groups were randomly placed in a separate hatcher unit. Those treatments included a non-injected control and the three 6/85 MG doses injected into the AC or AM. On each of 7 tray levels, eggs belonging to one treatment group were placed in 4 sections in each of 2 hatching baskets (8 total replicate sections; 30 eggs in each replicate section). To eliminate cross-contamination via chick droppings, hatching baskets containing eggs that had received the high (1.73 × 10^4^ CFU) 6/85 MG dosage were stacked at the bottom of the hatcher, with the medium dose (1.73 × 10^2^ CFU) eggs in the middle, and the low dose (1.73 CFU) eggs at the top portion of the hatcher. The hatching baskets containing the non-injected control eggs were placed above the hatching baskets containing the low dose eggs (Table 2).

A group of 240 eggs belonging to each of 2 treatment groups were randomly placed in another separate hatcher unit. Those treatments included eggs that received diluent injections into either the AC or AM. With 2 treatment groups assigned to each tray level, eggs from one treatment group were randomly allocated to one of two hatching baskets on each of 8 replicate tray levels (30 eggs in each replicate hatching basket) to coincide with the treatment replicate groups in the setter. Diluent-injected control (AC and AM) eggs were incubated in a separate hatcher unit from that containing the non-injected and 6/85 MG-injected eggs in order to prevent possible 6/85 MG cross-contamination of the diluent-injected control eggs (Table 3). Therefore, the data for the AC and AM diluent-injected treatment groups were analyzed separately.

A total of 64 eggs (one egg from each injection treatment group on each replicate tray level of each incubator) were injected with Coomassie brilliant blue R-250 dye (Genlantis, San Diego, CA, USA) and opened to locate the site of injection including the AC, AM, and body proper [54]. Hatch success and residual embryonic mortality were determined at 22 doi according to the procedures described by Alqhtani et al. [31], and Mousstaaid et al. [55,56]. In order to inject eggs into the AC site, plastic gapping sleeves were placed on all injector machine needles. The 6/86MG vaccine was applied in the AC of eggs in sequence from the lowest to highest dosage, and a cleaning cycle was applied between each individual dosage treatment. The gapping plastic sleeves were then removed and a full cleaning cycle was run before applying the 6/85 MG vaccine in the AM treatment group. In the AM treatment group, the 6/85 MG vaccine was likewise injected sequentially from the lowest to highest dosage. In the incubator in which both non-injected and the 6/85 MG-injected eggs were incubated together, the non-injected treatment group was individually compared with the site of injection–6/85 MG dosage combination treatment group. Fatemi et al. [45,46,47] have shown that the hatch results of non-injected control eggs do not differ from diluent-injected controls. Therefore, it is suggested that the non-injected and diluent-injected controls in this study would be comparable to each site of injection–6/85 MG dosage treatment combination.

Both hatchers were located in the same room and were set at the same temperature and humidity (36.7 °C dry bulb and 28.3 °C wet bulb (55% RH)) for the entire hatcher period. Similar to conditions in the setter, the temperature and humidity of both hatcher units were monitored by HOBO loggers to insure an accurate reading of the environmental conditions. The incubators and data loggers were monitored daily to ensure their proper function and that the eggs were incubated under optimum conditions.

### 2.3. Sampling at Hatch

All hatched chicks were pulled at 22 doi (526 h of incubation) and hatched chicks were counted, feather sexed and weighed, and only females were used in the posthatch phase of the study. Residue eggs were marked, counted, and subsequently opened for embryonic development stage confirmation [57]. Percentage egg weight loss (PEWL) in the 0 to 12, 12 to 18, and 0 to 18 doi intervals, hatchability of the injected eggs containing viable embryos (**HI**), and hatchling body weight (**BW**) at 22 doi were measured. Percentage egg weight loss was calculated based on the difference between the average initial and final weights of each of the groups of eggs in each of the 8 replicate flats in each treatment within each doi interval. Because of the removal of non-live embryonated eggs at 12 doi, the 12 to 18 doi PEWL is representative of only live embryonated eggs. Hatchery residue analysis variables at 22 doi included pre-pipped, pipped, and hatched chick mortalities. At hatch (22 doi), pre-pipped, pipped, and hatched chick mortalities were identified, respectively, as embryos that died prior to pipping externally (eggshell penetration), embryos that pipped externally but did not fully hatch, and chicks that fully hatched but were dead at time of pull. The hatchability of live embryonated eggs (HI) included those eggs that contained embryos that eventually hatched successfully and were alive at time of pull (22 doi). In each of the 8 replicate groups for each treatment, 25 chicks were weighed for determination of hatchling BW.

At hatch, a subset of 8 chicks were randomly chosen from each replicate–treatment group (72 total chicks), and were then euthanized, and their choanal clefts were immediately swabbed for 6/85 MG detection using pre-wetted sterile swabs in phosphate-buffered saline. Controls and the 1.73, 1.73 × 10^2^, and 1.73 × 10^4^ CFU dosage treatment groups were swabbed in that order. The MG detection test using real-time PCR was performed according to the procedure described by Elliott et al. [29] and Elliott et al. [38].

### 2.4. Posthatch Bird Raising and Sampling

Chicks from each replicate group in each treatment were pooled. The chicks were raised to 3 weeks of age in suspended battery cages that were housed in one room. Each battery cage measured 0.76 m × 0.46 m (0.35 m^2^). From a pool of 216 chicks belonging to the non-injected, diluent-injected, and MG-vaccinated treatment groups, 6 were randomly selected and placed in each of 4 replicate cages belonging to each of the 9 treatment groups. For the growing phase each of 9 treatment combinations had 4 replicates which accommodated the design of the facility used for this study. The stocking density of the 6 chicks in each cage was 0.06 m^2^ per bird to meet Hy-Line W-36 breeder pullet recommendations (Hy-Line Red Book, Hy-Line International, 2014). All birds had free access to water and feed during the grow-out period. The pullets were fed a crumble starter diet that met the recommended NRC requirements [58]. Animal brooding was performed according to Hy-Line recommendations for W-36 pullets (Hy-Line, 2020). Birds belonging to the control treatments were monitored daily before those belonging to the 6/85 MG treatments in order to prevent cross-contamination. Birds in the 1.73, 1.73 × 10^2^, and 1.73 ×10^4^ CFU dosage treatment groups were monitored daily, in that respective order, to further prevent cross-contamination. Chick mortality was monitored daily and dead chicks were weighed on a daily basis. Mean bird BW for each treatment replicate cage was measured on d 0, 7, 14, and 21 posthatch, and BW gain (**BWG**) was determined in the 0 to 7, 7 to 14, 14 to 21, and 0 to 21 d intervals. At 3 weeks of age, all birds were swabbed in the choanal cleft to test for the presence of 6/85 MG. The protocol by Nascimento et al. [59] was used for 6/85 MG detection by PCR.

### 2.5. Blood Sampling and Immunology

After the birds were swabbed at 3 wk of age, they were immediately bled. Immunological assessments, including serum plate agglutination (**SPA**) for the presence of IgM antibodies and ELISA for the presence of IgG antibodies against MG, were measured according to the methods described by Elliott et al. [30]. The ELISA test was only performed for treatments that were positive by SPA or when MG DNA was detected. Treatments that tested negative by SPA or for MG-DNA at hatch were not tested by ELISA for the presence of IgG antibodies against MG. Those treatments that tested negative for MG DNA presence included non-injected; diluent-injected in the AM and AC; and the 1.73, 1.73 × 10^2^, and 1.73 ×10^4^ CFU dosages injected in the AC.

### 2.6. Statistical Analysis

A randomized complete block design was employed for the hatch results, which included HI, PEWL, egg residue analyses, and hatchling BW, where incubator tray level was considered as the blocking factor. All treatments were randomly assigned to each block (tray level). Those treatments housed in the same incubator were compared statistically. Statistical comparison of the AC and AM diluent-injected control treatments were analyzed separately from the non-injected and 6/85 MG injection treatments, as they were housed in separate incubators. Posthatch BW, BWG, mortality, MG DNA detection by real-time PCR, and MG antibodies detected by SPA and ELISA tests were analyzed using a completely randomized experimental design, where replicate cage was the experimental unit. A 3 dosage (1.73, 1.73 × 10^2^, and 1.73 ×10^4^ CFU of 6/85 MG) × 2 site (AC and AM) of injection factorial arrangement of treatment was used for the hatch data of the 6/85 MG treatment groups housed in the same incubator and for the posthatch results of the 6/85 MG treatment groups in the grow-out facility. A one-way ANOVA was used to compare the AC and AM diluent-injected treatments and to individually compare the non-injected treatment group with each of the 6/85 MG treatment groups. All variables were analyzed by SAS 9.4 [60] employing PROC MIXED, and means separations were performed using Tukey’s protected least significant difference. Statements of significance were based on *p* ≤ 0.05 unless otherwise stated.

## 3. Results

### 3.1. Hatch Variables

#### 3.1.1. Comparison of Site of Injection (AC and AM) Treatments in the Diluent-Control Treatment Group

All eight eggs from each of the treatments that were tested for injection site accuracy with the use of dye were successfully injected in the intended site (AC or AM). In the diluent-injected controls, there were no significant differences due to site of injection (AC and AM) treatment for PEWL in any of the time intervals; HI; hatchling BW; pre-pipped and pipped embryonic mortalities; and hatched chick mortality. These results indicate that there were no significant effects due to the site of injection on the hatch results when only diluent was applied.

#### 3.1.2. Comparison of the Non-Injected Control Treatment Group with the Individual Site of Injection-6/85 MG Dosage Combination Treatment Groups

There were no significant differences between non-injected controls and each site of injection–6/85 MG dosage treatment combination for PEWL in each of the three intervals; hatchling BW; or for post-injection pre-pipped embryonic mortality. However, HI was decreased, and pipped embryo and hatched chick mortalities were increased in the 1.73 × 10^4^ CFU-AM treatment combination as compared with all other treatment combinations (Table 4).

#### 3.1.3. Effects of Site of Injection and 6/85 MG Dosage Treatments during the Incubation Period

For the eggs injected with 6/85 MG, there were no significant main or interactive effects due to dosage or site of injection treatment for PEWL for any time interval, nor for hatchling BW. However, there was a significant site of injection × dosage interactive effect on HI, where chicks in the 1.73 × 10^4^ CFU-AM treatment combination experienced a significantly lower (approximately 13% or more) HI in comparison with all the other treatment group combinations (Table 5).

There were also no interactive effects involving 6/85 MG dosage for post-injection pre-pipped (18 to 22 doi) embryo mortality, but pre-pipped embryo mortality was significantly increased when eggs were injected in the AM rather than the AC. Additionally, there was a notable trend that approached significance (*p* = 0.058) concerning the effects of *in ovo* treatment on pre-pipped embryo mortality, in which the high dose (1.73 × 10^4^ CFU) tended to increase pre-pipped embryo mortalities in comparison with the medium (1.73 × 10^2^ CFU) dose (Table 4). Hatched chick mortality was significantly increased when 1.73 × 10^4^ CFU of 6/85 MG was injected in the AM as compared with all other treatment combinations (Table 6).

### 3.2. Posthatch Variables

#### 3.2.1. Comparison of Site of Injection (AC and AM) Treatments in the Diluent-Control Treatment Group

In diluent-injected controls, there were no significant differences due to site of injection (AC or AM) for BW at 0, 7, or 14 d posthatch. However, BW on d 21 posthatch was significantly higher in the AM in comparison with the AC treatment. There were also no significant differences for BWG in the 0 to 7, 7 to 14, 14 to 21, and 0 to 21 d intervals, and for total bird mortality between 0 and 21 d posthatch (Table 6). However, there was a notable trend that approached significance concerning the effects of *in ovo* treatment on 14 to 21 d BWG (*p* = 0.057), in which the AM treatment groups tended to have a higher BWG than the AC treatment group (Table 7).

#### 3.2.2. Comparison of the Non-Injected Control Group with the Individual Site of Injection-6/85 MG Dosage Combination Treatment Groups

There were no significant differences between non-injected controls and each site of injection–6/85 MG dosage treatment combination for BW or BWG throughout the growing phase. However, total chick mortality through d 21 posthatch was significantly higher in the 1.73 × 10^4^ CFU-AM treatment combination in comparison with non-injected controls (Table 8).

#### 3.2.3. Main and Interaction Effects of Site of Injection and 6/85 MG Dosage Treatments

There were no significant main or interaction effects due to site of injection and 6/85 MG dosage on BW and BWG at any of the specified times during the grow out period. However, there was a numerical difference between the AC and AM treatment groups (*p* = 0.074) for BW at d 21, wherein the AM treatment group tended to have a higher BW on d 21 as compared with the AC treatment group (Table 9). Furthermore, there was a significant site of injection ×6/85 MG dosage interaction for total chick mortality through d 21 posthatch in the current study (Table 9).

#### 3.2.4. Hatch and Posthatch DNA Detection and Immune Response to 6/85 MG Treatment

The PCR results of the choanal cleft swabs taken from chicks belonging to the non-injected control group at hatch and at 3 wk posthatch were negative for MG. The serum samples taken at 3 wk posthatch from that same control group for SPA analysis also tested negative for IgM against MG. Birds in the diluent-injected group showed similar PCR and SPA results to those of the non-injected control group. The main and interaction effect means due to injection dosage and site for the PCR results of the choanal cleft swabs taken at hatch and at 3 wk posthatch, and the IgM and IgG serologic responses at 3 wk of age are shown in Table 8. The percentage of birds that tested positive for the presence of MG DNA was significantly higher in the AM treatment groups in comparison with the AC groups. In addition, at hatch, the percentage of MG DNA was significantly higher in the 1.73 × 10^4^ CFU treatment, lower in the 1.73 CFU treatment, and intermediate in the 1.73 × 10^2^ CFU treatment. Furthermore, birds in the 1.73 × 10^2^ CFU-AM and 1.73 × 10^4^ CFU-AM treatment combination tested positive for the presence of MG DNA as compared with all treatment combination groups at 3 wk posthatch. The presence of MG DNA was also higher in the 1.73 × 10^4^ CFU-AM treatment combination in comparison with the 1.73 × 10^2^ CFU-AM treatment combination at 3 wk posthatch. The levels of IgM were significantly higher in the 1.73 × 10^2^ CFU-AM and 1.73 × 10^4^ CFU-AM treatment combination groups in comparison with all other treatment combination groups at 3 wk posthatch. Moreover, IgG levels were undetectable in the AC-treatment groups at 3 wk posthatch. There was also a noticeable trend (*p* = 0.068) in IgG levels at 3 wk posthatch due to the site of injection and 6/85 MG dosage, in which the 1.73 × 10^4^ CFU-AM combination treatment tended to have higher IgG levels than those of the 1.73 CFU-AM and 1.73 × 10^2^ CFU-AM treatment combinations (Table 10).

## 4. Discussion

The aim of the current study was to determine the effects of various dosages of the *in ovo* injection of 6/85 MG vaccine on the hatching process, early live performance, and serological response of layers. *Mycoplasma gallisepticum* is the organism that causes avian respiratory mycoplasmosis, leading to chronic respiratory disease in chickens, and is responsible for reductions in egg production and other economic losses in the poultry industry [8]. Mycoplasmosis in layer pullets has also been shown to result in impaired live performance [39] as well as a depressed immune response [61]. Effects of the *in ovo* administration of FMG [29,30] and ts-11MG [31] on hatchling quality and early live performance have been previously tested. However, the *in ovo* injection of medium to high dosages of FMG has had a detrimental effect on the hatching process and posthatch livability of pullets [29,30]. Additionally, ts-11 MG has been demonstrated not to show any immune protection at any dosage of injection in hatchlings and early posthatch pullets [31]. The results of the current study confirm that the 1.73 × 10^4^ CFU-AM (high dose in the amnion) treatment combination has a proven negative impact on the normal hatching process as well as early posthatch livability. These results are partially in agreement with previous FMG *in ovo* injection studies. In this current study, the HI, pipped embryonic mortality, and hatched chick mortality rates in the 1.73 × 10^4^ CFU-AM treatment group were 82.78%, 2.70%, and 6.85%, respectively. However, in the study by Elliott et al. [29], the HI of eggs injected *in ovo* with 2.4 × 10^4^ CFU of FMG was 75.00%. Furthermore, Elliott et al. [29] have reported that the pipped embryo mortality and hatched chick mortality rates in the 2.4 × 10^4^ CFU treatment group were 16.7% and 25.0%, respectively. The inconsistent results between the study conducted by Elliott et al. [29] and this study could be linked to the different virulence levels of the MG strains. It is well documented that the FMG vaccine is more virulent than the ts-11 or 6/85 MG strains [13]. Therefore, the differing results could be attributed to the greater virulence of FMG compared with that of 6/85 MG.

By 8 doi of the 21 doi period of embryogenesis, the first signs of immune system activity are observed in the immune organs including the bursa and spleen [62]. On 11 and 12 doi, T cells and B cells begin to develop respectively, with B cell differentiation primarily occurring after 15 doi. By 18 doi, the chicken embryo displays immunocompetence and is capable of producing both an innate and an adaptive response to pathogens [63,64]. However, the negative effects of a high dosage 6/85 MG *in ovo* injection on embryo and chick survival could be linked to the way that the immune system is relatively immature during the incubation period. It is well documented that the immune response of chicken embryos begins to develop after the first week of incubation, with full development being completed during the first 10 days of posthatch age [65]. Thus, chicken embryos are more susceptible to enteric pathogenic agents during the incubational and early posthatch periods. Importantly, immune responses have been shown to be comparable in birds either vaccinated *in ovo* or at a later time during the posthatch period [36]. Furthermore, *in ovo* administration of the MDV in broilers has been shown to increase the proportions of T cell subsets in the spleen [66]. Increasing the splenic expression of type I IFNs and TLRs leads to early maturation of the immune response. Likewise, an appropriate humoral immune response has been observed in six-week-old pullets against MG when they received an *in ovo* injection of FMG ranging from 2.4 × 10^2^ to 2.4 × 10^6^ CFU at 18 doi [30]. Similar to previous studies, the *in ovo* injection of 6/85 MG ranging from 1.73 × 10^2^ CFU to 1.73 × 10^4^ CFU (medium to high dosage) into the AM stimulated a humoral immune response at 3 wk posthatch. These results indicate that the *in ovo* injection of MG can trigger an immune response in chicken embryos and result in the early development of their immune system.

The results of the current study agree with those of Levisohn et al. [67] and Viscione et al. [24], who observed no negative effects of the gavage-inoculation of 10^2^ CFU of FMG and 6/85 MG, respectively, on layer pullet BW and BWG at 10 wk posthatch. Upon consideration of hatch quality, live performance, and immune protection, the administration of a 1.73 × 10^2^ CFU dose of 6/85 MG in the AM was safer than the AM administration of the same dose of FMG. Furthermore, various birds from the 1.73 × 10^4^ CFU-AM treatment group exhibited clinical signs of extended and twisted neck breathing (Figure 1).

The 1.73 × 10^4^ CFU-AM treatment may have affected the tracheal and lung functions of the chicks due to its increased level of colonization, and a subsequent increase in respiratory failure. The results of Viscione et al. [24] and Levisohn et al. [67] are similar concerning the effects of a pre-lay inoculation of FMG on subsequent BW and show that, despite differences in bird type and the doses and timing of FMG inoculation, it has minimal effects on BW in the early phases of growth as well as after sexual maturity. Therefore, it is recommended that when 6/85 MG injections are given in the AM, the doses should be less than 1.73 × 10^4^ CFU. No chick mortalities were observed in the non-injected control group. Fatemi et al. [47] showed that the posthatch performance of broiler chicks hatched from non-injected control eggs was not different from those hatched from diluent-injected control eggs. Based on those previous results, it is suggested that the live performance of broiler chicks hatched from eggs belonging to the non-injected control group would also not be significantly different from those hatched from eggs belonging to the diluent-control group in this study. The posthatch results of non-injected controls were not different from those in all of the sites of injection–6/85 MG dosage combination treatment groups, except for the 1.73 × 10^4^ CFU-AM treatment, which indicates that the injection of the 1.73 or 1.73 × 10^2^ CFU doses in the AM are safe. Although the AC site was safe at all the doses, it did not allow for transmission of the bacteria to the embryos. These results confirm that although the 1.73 × 10^4^ CFU-AM treatment combination did not affect the growth of viable chicks, it has a proven negative impact on chick livability through 21 d posthatch.

In addition, besides the effects of the administered dosage of 6/85 MG, differences were observed concerning the effects of AC and AM injection on posthatch performance, humoral immune response, and MG DNA detection. When injected in the AC, chicks exhibited no immune protection at 3 wk posthatch, and had a lower BW and BWG between 0 and 21 d of posthatch. Furthermore, MG DNA was only detectable at hatch and it remained undetectable after hatch, indicating that injection in the AC site did not allow for bacteria to be successfully transmitted to the embryo. Williams [42] and Williams [43] have reported that the *in ovo* vaccination of broilers with MDV in the AM resulted in approximately 90% immune protection while AC injections elicited no immune protection. This indicates that, in comparison with AM administration, injection materials in the AC are less efficiently assimilated. It has been well reported that the *in ovo* injection of diluent in the AM alone can be beneficial to the hatchability of fertile eggs and to hatchling BW [44]. Furthermore, the *in ovo* injection of MDV in combination with diluent has been shown to increase the villus length to crypt depth ratio in the jejunum when compared with field vaccinated broilers [28,68]. An increase in villus length to crypt depth ratio is associated with an increased BW and BWG [45,46,48,69]. Therefore, improvements in the serological response as well as the live performance of pullets injected in the AM would be associated with a better absorption of vaccines suspended in diluent.

## 5. Conclusions

In conclusion, effects of the *in ovo* injection of 6/85 MG in the AM and AC on HI, hatch residue, posthatch performance, and bird serology were investigated. The current findings reveal that the 6/85 MG vaccine caused no posthatch chick and posthatch mortality when injected into the AC at all dosages and in the AM at the 1.73 and 1.73 × 10^2^ CFU dosages. However, a 1.73 × 10^4^ CFU dosage level of 6/85 MG resulted in greater embryo and early-posthatch mortalities when injected in the AM. The AM site of injection was found to be more effective compared with the AC site which was confirmed by the MG DNA, IgM, and IgG results when 6/85 MG was injected at 18 doi. These results indicate that the 1.73 × 10^2^ CFU treatment injected in the AM would be the best candidate for pragmatic commercial application, in that it did not lead to embryo and posthatch chick mortalities, but at the same time elicited a humoral response. An MG field challenge study is needed to further evaluate the effectiveness of an *in ovo* 6/85 MG vaccination and its capability to provide protection against field strain MG infections in the pre-lay and lay periods of commercial flocks.

## Figures and Tables

**Figure 1 animals-13-01228-f001:**
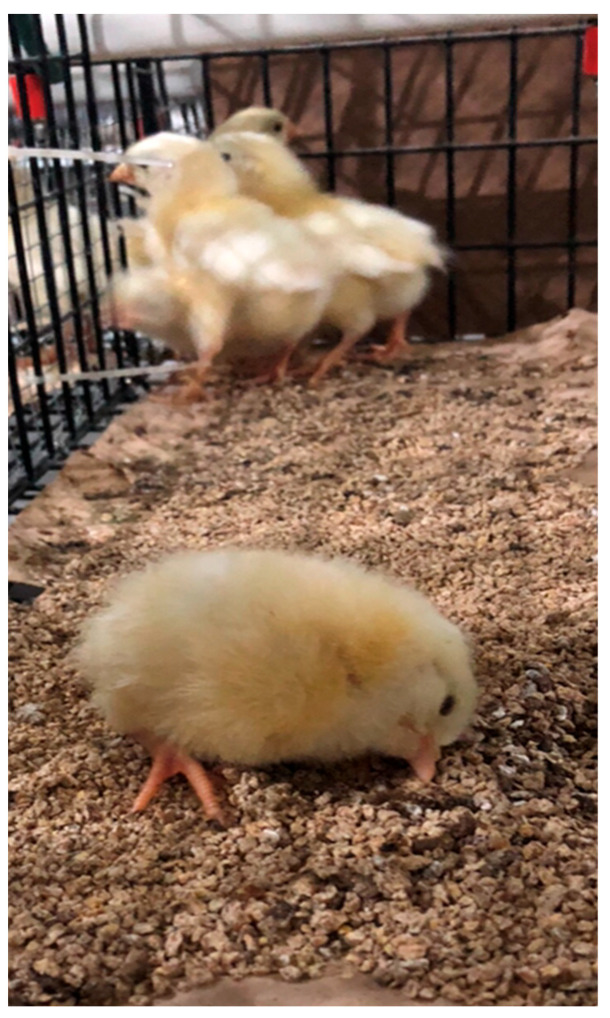
Hy-Line W-36 layer chicks exhibiting twisted and extended necks after receiving the high dose (1.73 × 10^4^ CFU) of 6/85 MG in the AM at 18 days of incubation. Chicks in all other treatment groups, including controls and the lower 6/85 MG treatment doses, did not exhibit this behavior.

**Table 1 animals-13-01228-t001:** Treatment dilutions and corresponding mean colony forming units (CFU) dose of 6/85 Mycoplasma gallisepticum for each 50 μL solution volume administered by *in ovo* injection to layer embryos at 18 days of incubation.

Treatment (Dilution)	Dose (CFU)
Low (1 × 10^−6^)	1.73
Medium (1 × 10^−4^)	1.73 × 10^2^
High (1 × 10^−2^)	1.73 × 10^4^

**Table 2 animals-13-01228-t002:** Experimental design in the hatchery unit containing eggs belonging to the following treatments: the non-injected control, 1.73, 1.73 × 10^2^, and 1.73 × 10^4^ 6/85 MG CFU doses injected into the amnion (AM) or air cell (AC).

Treatment Tray Level	Replicate Hatching Basket Sections ^1^	Treatment
1	8	Non-injected
2	8	1.73 CFU-AM
3	8	1.73 CFU-AC
4	8	1.73 × 10^2^ CFU-AM
5	8	1.73 × 10^2^ CFU-AC
6	8	1.73 × 10^4^ CFU-AM
7	8	1.73 × 10^4^ CFU-AC

^1^ Thirty eggs were randomly assigned to eight replicate hatching basket sections that belonged to the designated treatment on one of seven treatment tray levels.

**Table 3 animals-13-01228-t003:** Experimental design in the hatchery unit containing eggs belonging to treatments in which diluent was injected into either amnion (**AM**) or air cell (**AC**).

Replicate Tray Level	Hatching Basket	Treatment ^1^
1	1	Diluent in the AM
	2	Diluent in the AC
2	1	Diluent in the AM
	2	Diluent in the AC
3	1	Diluent in the AM
	2	Diluent in the AC
4	1	Diluent in the AM
	2	Diluent in the AC
5	1	Diluent in the AM
	2	Diluent in the AC
6	1	Diluent in the AM
	2	Diluent in the AC
7	1	Diluent in the AM
	2	Diluent in the AC
8	1	Diluent in the AM
	2	Diluent in the AC

^1^ Thirty eggs were randomly assigned to a hatching basket that belonged to the designated treatment on each of eight replicate tray levels.

**Table 4 animals-13-01228-t004:** Percentage egg weight loss (**PEWL**) between 0 and 12, 12 and 18, and 0 and 18 days (**d**) of incubation (**doi**) and hatchability of injected eggs containing viable embryos (**HI**), mean hatchling BW, pre-pipped and pipped embryonic mortality, and hatched chick mortality determined at pull time (22 doi) in non-injected eggs or eggs injected in the air cell (**AC)** or amnion (**AM**) with various 6/85 MG dosages (1.73, 1.73 × 10^2^, and 1.73 × 10^4^ CFU).

	PEWL	PEWL	PEWL	HI ^1^	Pre-Pipped Embryo Mortality ^1^	Pipped Embryo Mortality ^1^	Hatched Chick Mortality ^1^	Hatchling BW ^2^
	0–12 d ^1^	12–18 d ^1^	0–18 d ^1^					
Dosage-Site of Injection	---------------------------------------------(%)---------------------------------------------							(g)
Non-injected	7.3	3.96	11.3	92.6 ^a^	4.84	1.39 ^b^	0 ^b^	34.2
1.73 CFU-AC	7.3	4.18	11.4	94.9 ^a^	3.34	1.43 ^b^	0.48 ^b^	34.7
1.73 × 10^2^ CFU-AC	7.3	4.16	11.5	94.9 ^a^	3.50	1.33 ^b^	0 ^b^	34.8
1.73 × 10^4^ CFU-AC	7.2	4.05	11.3	95.4 ^a^	3.64	0.95 ^b^	0 ^b^	34.8
1.73 CFU-AM	7.4	4.15	11.5	91.7 ^a^	3.90	2.88 ^ab^	0 ^b^	34.5
1.73 × 10^2^ CFU-AM	7.2	4.05	11.3	95.4 ^a^	3.64	0.95 ^b^	0 ^b^	34.8
1.73 × 10^4^ CFU-AM	7.2	4.09	11.2	83.8 ^b^	5.53	3.61 ^a^	6.84 ^a^	34.9
Pooled SEM	0.12	0.091	0.19	2.92	2.316	1.712	1.447	0.44
Interaction *p*-value	0.518	0.254	0.772	0.005	0.910	0.030	0.001	0.780

^a,b^ Treatment means within the same variable column within the type of treatment with no common superscript differ significantly (*p* < 0.05). ^1^ N = 30 eggs in each of 8 replicate groups (trays) in each of 7 treatments were used for means calculations. ^2^ N = 25 chicks in each of 8 replicate groups (hatching basket sections) in each of 7 treatments were used for means calculations.

**Table 5 animals-13-01228-t005:** Main and interaction means of percentage egg weight loss (**PEWL**) between 0 and 12, 12 and 18, and 0 and 18 days (**d**) of incubation (**doi**) and hatchability of injected eggs containing viable embryos (**HI**) and mean hatchling BW determined at pull time (22 doi) due to 6/85 MG dosage (1.73, 1.73 × 10^2^, and 1.73 × 10^4^ CFU) and site of injection (air cell (**AC**) or amnion (**AM**)).

	PEWL 0–12 d ^1^	PEWL 12–18 d ^1^	PEWL 0–18 d ^1^	HI ^1^	Hatchling BW ^2^
	--------------------------(%)--------------------				(g)
Dosage					
1.73 CFU^2^	7.24	4.11	11.34	92.41 ^a^	34.78
1.73 × 10^2^ CFU^3^	7.23	4.10	11.35	95.14 ^a^	34.74
1.73 × 10^4^ CFU^4^	7.29	4.14	11.43	88.84 ^b^	34.79
Pooled SEM	0.070	0.056	0.127	2.030	0.277
Main effect *p*-value	0.671	0.741	0.723	0.014	0.978
**Site of Injection**					
AC	7.28	4.12	11.38	94.31 ^a^	34.67
AM	7.23	4.12	11.37	89.94 ^b^	34.87
Pooled SEM	0.064	0.049	0.105	1.677	0.222
Main effect *p*-value	0.427	0.944	0.908	0.014	0.372
**Dosage-Site of Injection**					
1.73 CFU-AC	7.35	4.13	11.46	92.67 ^a^	34.64
1.73 × 10^2^ CFU-AC	7.19	4.05	11.25	95.38 ^a^	34.68
1.73 × 10^4^ CFU-AC	7.31	4.18	11.44	94.90 ^a^	34.69
1.73 CFU-AM	7.14	4.09	11.23	92.15 ^a^	34.92
1.73 × 10^2^ CFU-AM	7.28	4.16	11.45	94.90 ^a^	34.80
1.73 × 10^4^ CFU-AM	7.28	4.12	11.43	82.78 ^b^	34.90
Pooled SEM	0.109	0.084	0.181	2.871	0.375
Interaction *p*-value	0.171	0.296	0.252	0.009	0.961

^a,b^ Treatment means within the same variable column within the type of treatment with no common superscript differ significantly (*p* < 0.05). ^1^ N = 30 eggs in each of 8 replicate groups (trays) in each dosage-injection site treatment combination were used for means calculations. ^2^ N = 25 chicks in each of 8 replicate groups (hatching basket sections) in each dosage-injection site treatment combination were used for means calculations.

**Table 6 animals-13-01228-t006:** Main and interaction means of post-injection pre-pipped and pipped embryonic mortalities, and hatched chick mortality determined at pull time (22 days of incubation) ^1^ due to 6/85 MG dosage (1.73, 1.73 × 10^2^, and 1.73 × 10^4^ CFU) and site of injection (air cell (**AC**) or amnion (**AM**)).

	Pre-Pipped Embryo Mortality	Pipped Embryo Mortality	Hatched Chick Mortality
	--------------------------------(%)------------------------------		
Dosage			
1.73 CFU	3.12	4.63	0.00 ^b^
1.73 × 10^2^ CFU	1.14	3.57	0.00 ^b^
1.73 × 10^4^ CFU	4.13	3.02	3.66 ^a^
Pooled SEM	1.221	1.594	1.102
Main effect *p*-value	0.058	0.729	0.002
**Injection Site**			
AC	1.668 ^b^	3.57	0.158 ^b^
AM	3.921 ^a^	3.91	2.283 ^a^
Pooled SEM	1.009	1.294	0.899
Main effect *p*-value	0.032	0.797	0.024
**Dosage-Injection Site**			
1.73 CFU-AC	2.63	3.74	0.00 ^b^
1.73 × 10^2^ CFU-AC	0.95	3.64	0.00 ^b^
1.73 × 10^4^ CFU-AC	1.43	3.34	0.00 ^b^
1.73 CFU-AM	3.61	5.53	0.00 ^b^
1.73 × 10^2^ CFU-AM	1.32	3.50	0.00 ^b^
1.73 × 10^4^ CFU-AM	6.83	2.70	6.85 ^a^
Pooled SEM	1.727	2.214	1.559
Interaction *p*-value	0.097	0.729	0.008

^a,b^ Treatment means within the same variable column within the type of treatment with no common superscript differ significantly (*p* < 0.05). ^1^ N = 30 eggs in each of 8 replicate groups (trays) in each dosage-injection site treatment combination were used for means calculations.

**Table 7 animals-13-01228-t007:** Means of body weight (**BW**) at 0 (**BW0**), 7 (**BW7**), 14 (**BW14**), and 21 (**BW21**) days (**d**) posthatch; BW gain (**BWG**) between 0 and 7 (**BWG0-7**), 7 and 14 (**BWG7-14**), 14 and 21 (**BWG14-21**) and 0 and 21 (**BWG0-21**) d posthatch; and total chick mortality through d 21 posthatch ^1^ due to injection site (air cell (**AC**) or amnion (**AM**)).

	BW (g)						BWG (g)		Mortality
	d0	d7	d14	d21	0–7 d	8–14 d	15–21 d	0–21 d	0–21 d
AC	35.3	62.6	111	176 ^b^	22.3	45.8	66.0	136.0	16.5
AM	35.5	62.9	117	186 ^a^	28.3	53.0	69.0	150.0	0.00
SEM	0.56	7.39	3.1	4.0	3.82	6.41	1.41	5.9	9.35
*p*-value	0.670	0.969	0.102	0.046	0.168	0.301	0.057	0.078	0.134

^a,b^ Treatment means within the same variable column within the type of treatment with no common superscript differ significantly (*p* < 0.05). ^1^ N = Six birds in each of four replicate groups in each dosage-injection site treatment combination were used for means calculations.

**Table 8 animals-13-01228-t008:** Means of body weight (**BW**) at 0 (**BW**0), 7 (**BW7**), 14 (**BW14**), and 21 (**BW21**) days (**d**) posthatch; BW gain (**BWG**) between 0 and 7 (**BWG0-7**), 7 and 14 (**BWG7-14**), 14 and 21 (**BWG14-21**) and 0 and 21 (**BWG0-21**) d posthatch; and total chick mortality through d 21 posthatch^1^ in non-injected eggs or eggs injected in the air cell (**AC**) or amnion (**AM**) with various 6/85 MG dosages (1.73, 1.73 × 10^2^, and 1.73 × 10^4^ CFU).

	BW (g)				BWG (g)				Mortality
	d0	d7	d14	d21	0–7 d	8–14 d	15–21 d	0–21 d	0–21 d
Dosage-Site of Injection									
Non-injected	34.0	57.8	115.8	179.8	23.5	59.3	63.8	143.8	0 ^b^
1.73 CFU-AC	34.8	61.7	113.3	170.5	27.0	51.5	57.3	135.8	0 ^b^
1.73 CFU-AM	34.0	57.5	110.8	189.5	22.8	53.3	73.3	149.4	0 ^b^
1.73 × 10^2^ CFU-AC	35.0	60.8	113.3	177.0	20.5	60.0	63.5	144.0	0 ^b^
1.73 × 10^2^ CFU-AM	34.3	62.4	112.0	174.0	29.0	48.5	61.8	139.3	0 ^b^
1.73 × 10^4^ CFU-AC	35.0	60.6	109.3	177.8	24.8	49.5	68.3	142.6	0 ^b^
1.73 × 10^4^ CFU-AM	34.3	65.9	117.8	187.5	29.5	53.3	70.0	132.8	41.5 ^a^
Pooled SEM	0.52	6.00	3.54	9.31	4.51	6.52	7.33	9.76	9.24
Interaction *p*-value	0.223	0.841	0.286	0.398	0.392	0.485	0.397	0.384	0.001

^a,b^ Treatment means within the same variable column within the type of treatment with no common superscript differ significantly (*p* < 0.05). N = Six birds in each of four replicate groups in each dosage-injection site treatment combination were used for means calculations.

**Table 9 animals-13-01228-t009:** Main and interaction means of body weight (**BW**) at 0 (**BW0**), 7 (**BW7**), 14 (**BW14**), and 21 (**BW21**) days (**d**) posthatch; BW gain (**BWG**) between 0 and 7 (**BWG0-7**), 7 and 14 (**BWG7-14**), 14 and 21 (**BWG14-21**) and 0 and 21 (**BWG0-21**) d posthatch; and total chick mortality through d 21 posthatch ^1^ due to 6/85 MG dosage and injection site (air cell (**AC**) or amnion (**AM**)).

	BW0	BW7	BWG0-7	BW14	BWG7-14	BW21	BWG14-21	BWG0-21	Mortality
	-----------------------------------------------(g)------------------------------------------------								(%)
Dosage									
1.73 CFU	34.59	63.94	29.35	114.84	50.90	180.64	65.80	146.06	0.00 ^b^
1.73 × 10^2^ CFU	34.88	58.70	23.85	110.00	51.30	183.55	70.78	148.66	0.00 ^b^
1.73 × 10^4^ CFU	34.80	60.78	23.88	113.25	55.81	173.63	60.36	136.70	20.83 ^a^
Pooled SEM	0.364	4.235	3.400	2.451	4.616	6.913	5.569	6.882	7.079
Main effect *p*-value	0.719	0.475	0.204	0.161	0.510	0.358	0.202	0.216	0.012
**Injection Site**									
AC	34.63	61.59	26.98	111.43	49.82	173.92	62.48	139.28	0.00 ^b^
AM	34.88	60.68	24.40	113.97	55.53	184.62	68.82	148.34	13.88 ^a^
Pooled SEM	0.297	3.458	2.773	2.001	3.769	5.645	4.547	5.620	5.781
Main effect *p*-value	0.396	0.796	0.364	0.220	0.147	0.074	0.180	0.124	0.027
**Dosage-Injection Site**									
1.73 CFU-AC	34.23	63.43	29.20	112.00	48.57	173.78	61.75	139.55	0.00 ^b^
1.73 × 10^2^ CFU-AC	35.00	59.70	24.70	109.18	49.45	177.60	68.40	142.55	0.00 ^b^
1.73 × 10^4^ CFU-AC	34.65	61.65	27.05	113.10	51.43	170.38	57.28	135.73	0.00 ^b^
1.73 CFU-AM	34.95	64.45	29.20	117.67	53.23	187.50	69.57	152.57	0.00 ^b^
1.73 × 10^2^ CFU-AM	34.75	57.70	23.00	110.83	53.15	189.50	73.15	154.78	0.00 ^b^
1.73 × 10^4^ CFU-AM	34.95	59.90	20.70	113.40	60.20	176.88	63.45	137.68	41.65 ^a^
Pooled SEM	0.515	5.989	4.803	3.467	6.528	9.777	7.876	9.733	10.012
Interaction *p*-value	0.424	0.925	0.612	0.534	0.844	0.864	0.956	0.674	0.012

^a,b^ Treatment means within the same variable column within the type of treatment with no common superscript differ significantly (*p* < 0.05). ^1^ N = Six birds in each of four replicate groups in each dosage-injection site treatment combination were used for means calculations.

**Table 10 animals-13-01228-t010:** Main and interaction means for percentage of birds that tested positive for MG DNA in the choanal cleft at 22 days (**d**) of incubation (**doi**) (**Hatch DNA**) and at 3 wk posthatch (**3 wk DNA**) and IgM (**3 wk IgM**) and IgG (**3 wk IgG**) serologic responses in the serum at 3 wk posthatch ^1^ due to 6/85 MG dosage and site of injection (air cell (**AC**) or amnion (**AM**)).

	Hatch DNA ^2^	3 wk DNA ^3^	3 wk IgM ^3^	3 wk IgG ^3^(mg/dL)
	-----------------------------------------------(%)-----------------------------------------			
Dosage				
1.73 CFU	18.75 ^c^	0.00 ^b^	0.00 ^b^	105.92
1.73 × 10^2^ CFU	43.75 ^b^	23.91 ^a^	32.61 ^a^	108.39
1.73 × 10^4^ CFU	62.50 ^a^	39.29 ^a^	42.86 ^a^	169.11
Pooled SEM	14.430	5.590	5.177	29.870
Main effect *p*-value	0.015	≤0.0001	≤0.0001	0.068
**Injection Site**				
AC	16.67 ^b^	0.00 ^b^	0.00 ^b^	0.00 ^b^
AM	66.67 ^a^	42.13 ^a^	50.3 ^a^	255.61 ^a^
Pooled SEM	11.785	4.466	4.132	23.730
Main effect *p*-value	≤0.0001	≤0.0001	≤0.0001	≤0.0001
**Dosage-Injection Site**				
1.73 CFU-AC	0.00	0.00 ^c^	0.00 ^c^	0.00
1.73 × 10^2^ CFU-AC	12.50	0.00 ^c^	0.00 ^c^	0.00
1.73 × 10^4^ CFU-AC	37.50	0.00 ^c^	0.00 ^c^	0.00
1.73 CFU-AM	37.50	0.00 ^c^	0.00 ^c^	211.83
1.73 × 10^2^ CFU-AM	75.00	47.83 ^b^	65.22 ^b^	216.78
1.73 × 10^4^ CFU-AM	87.50	78.57 ^a^	85.71 ^a^	338.21
Pooled SEM	20.412	7.288	6.744	38.735
Interaction *p*-value	0.689	≤0.0001	≤0.0001	0.068

^a–c^ Treatment means within the same variable column within the type of treatment with no common superscript differ significantly (*p* < 0.05). ^1^ N = Six birds in each of four replicate groups in each dosage–injection site treatment combination were used for means calculations. ^2^ N = One bird in each of eight replicate flats in each dosage–injection site treatment combination were used for means calculations. ^3^ N = Six birds in each of four replicate groups in each dosage–injection site treatment combination were used for means calculations, except for the high dose AM treatment (14 chicks total) due to chick mortality.

## Data Availability

None of the data were deposited in an official repository.

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
