# Peer review of "Effects of the In ovo Administration of the 6/85 Mycoplasma gallisepticum Vaccine on Layer Chicken Embryo Hatchability and Early Posthatch Performance"

_animals, 2023, doi:10.3390/ani13071228_

Round 1

Reviewer 1 Report

Dear author

Your manuscript is significant; however, your writing needs to be improved substantially.

  Simple summary

You cite the HI in "Simple Summary"; however, in this section, you don't explain the meaning.

Abstract

The abstract isn't adequate because the methodology described in the abstract is confusing. The conclusion of "simple summary" is different from abstract

Introduction

Line 81 – Please separate the words "been" and "recommended"

Line 107 – I didn't understand why you compared L-ascorbic acid. Was it injected via amnio?

The 6/85 MG strain is avirulent, right? Put the history and/or information about this strain.

Mycoplamas gallisepticum is not a common disease in chicks (Young birds), so why is it important to vaccinate birds in ovo? Please explain it to the reader in the introduction.

Material and method

LIne 119 – the birds were vaccinated to MG? In my country, vaccinating breeding hens for MG is not allowed. However, we do not have this information.

Line 136 – Table 1 is not necessary.

Line 141 – how many eggs were removed at 12 doi?

LIne 153 – Please put this information clear to the reader.

Line 207 (item 2.4)  -  Were there 216 birds per group?

Line 234-235 – Please put this sentence clear to the reader

Results

Line 271-274  -  This sentence was confusing. Please make it clearer for the reader.

Tables (all tables) – Please put the meaning of abbreviations at the foot of the tables.

Table 2 – it's missing the group 173CFU-AM

Tables 3, 4, 7, 8 don't provide as much information when you separate dosage and place of inoculation. Therefore, I think just the information about 9 groups is necessary. If you disagree with me, please, explain why.

What's the difference between tables 2 and 3? What is new in table 3 in relation to table 2?

Do you need tables 6 and 7 to show your results?

Table 5 – what's AN?

Results titles are uninteresting to the reader. Example: "3.1.1. Comparison of site of injection (AC and AM) treatments in the diluent-control treatment group". You could use: "The site of vaccine inoculation (amnion or shell membrane) does not result in mortality or injury to the embryos". You could change all titles.

Discussion

As with the results title, I found your discussion to be very "conservative". You presented more about whether your work agrees or disagrees with the results of other authors who worked with the F strain than the discussion of your results.

Strain F is a virulent strain, so we can cite this strain but not compare it in all times.

For example: LIne 423, you cite Elliot and Branton and use the word "inconsistente": Your study is not inconsistent with this author. You just used an non virulent strain.

You should focus on the advantages of your results: You can use the strain 6/85 via AM in ovo at a dosis of 173 or 1,73x102 CFU/chicken embryo safely and with good antibody titers after 3 weeks of age.

I quoted the dose of 173 because in this dose we didn't have a lesion or dead chicken embryo, your number of MG was zero at 21 days age but with a title of antibody similar to dose of 1,73x102 and 1,73x104. So why did you not discuss the dose of 173 CFU?

Figure 1: I think figure 1 should be put in supplementary material.

Conclusion

Your conclusion looks like a discussion. The decision lacks objectivity.

You should use up 3 lines for the conclusion. For example:

The Mycoplasma gallisepticum strain 6/85 can be a secure and efficient vaccine in ovo when inoculated in fluid amnion at a dose of 1,73x102 CFU/chicken embryo.

Author Response

Reviewer 1:

Your manuscript is significant; however, your writing needs to be improved substantially.

  Simple summary

  1. You cite the HI in "Simple Summary"; however, in this section, you don't explain the meaning.

Answer:

The relevant changes were applied in this section. 

Abstract

  1. The abstract isn't adequate because the methodology described in the abstract is confusing. The conclusion of "simple summary" is different from abstract

Answer:

The relevant changes were applied in this section. 

Introduction

  1. Line 81 – Please separate the words "been" and "recommended"

Answer:

The relevant correction was applied. 

  1. Line 107 – I didn't understand why you compared L-ascorbic acid. Was it injected via amnio?

Answer:

Thank you for the comments. In that reference, the L-ascorbic acid was injected into the AC whereas the MDV was injected into the AM. The reason for the use of this reference was to demonstrate that the difference in the injected materials can be effective when they were injected into the different sites. The relevant corrections were applied in this section.

  1. The 6/85 MG strain is avirulent, right? Put the history and/or information about this strain.

Answer:

The 6/85 MG vaccine information is avirulent, but it has been shown that it can provide immunization against MG in layers without having negative effects on the egg production. Also, it is approved by USDA for commercial application. This information is presented in the current draft between lines 70 and 79.

  1. Mycoplamas gallisepticumis not a common disease in chicks (Young birds), so why is it important to vaccinate birds in ovo? Please explain it to the reader in the introduction.

 Answer:

Although it is not common in early posthatch pullets, it can be beneficial by providing early protection against MG and to subsequently lower MG infection during lay.

Material and method

  1. Line 119 – the birds were vaccinated to MG? In my country, vaccinating breeding hens for MG is not allowed. However, we do not have this information.

Answer:

In some areas of the US, pullets, and not broilers, were vaccinated against MG. Turkeys are susceptible to the FMG and it is not approved for use in some states where turkeys are grown. However, this sentence means that the eggs that were used in this study were collected from flocks that were not contaminated with MG. 

  1. Line 136 – Table 1 is not necessary.

Answer:

This table is used for better presentation of the in ovo treatments as well as showing the actual number of cells in each treatment group. Authors prefer to include this information.

  1. Line 141 – how many eggs were removed at 12 doi?

Answer:

The number of eggs that were removed during candling was not our concern. Because we incubated 20% more eggs than we expected for the growing phase and the percentage of eggs that were removed after candling was below 5%.

  1. Line 153 – Please put this information clear to the reader.

Answer:

The machine is designed to inject into the amnion. However, as is indicated, we used plastic gapping sleeves to adjust injection depth to provide air cell injections. The following sentence from the current draft explained the aforementioned process: “In order to inject eggs into the AC site, plastic gapping sleeves were placed on all injector machine needles.”.

  1. Line 207 (item 2.4)  -  Were there 216 birds per group?

Answer:

The total numbers of birds were 216. This number comes from 6 birds in each cage with 4 treatment replications for each of the 9 treatments.

  1. Line 234-235 – Please put this sentence clear to the reader

Answer:

 The relevant changes were performed in this sentence.

Results

  1. Line 271-274  -  This sentence was confusing. Please make it clearer for the reader.

Answer:

The relevant corrections were applied as follows:

“However, HI was decreased, and pipped embryo and hatched chick mortalities were increased in the 1.73 x ‎‎104 CFU-AM treatment combination as compared to all other treatment combinations“‎

  1. Tables (all tables) – Please put the meaning of abbreviations at the foot of the tables.

Answer:

Thank you for the comment. However, all relevant abbreviations were defined in either the title or footnotes of the table. “SEM” and “CFU” are general acceptable terms that were approved to be abbreviated by this journal. 

  1. Table 2 – it's missing the group 173CFU-AM

Answer:

The relevant correction was applied.

  1. Tables 3, 4, 7, 8 don't provide as much information when you separate dosage and place of inoculation. Therefore, I think just the information about 9 groups is necessary. If you disagree with me, please, explain why.

Answer:

Thank you for the comment. We had 2 control groups: non-injected and diluent-injected. However, we did not have any diluent-injected treatments that were incubated with those treatments that were injected with vaccine via either the amnion or air cell. Therefore, we cannot statistically compare their measurements with the vaccine-injected or non-injected treatments. Although we presented and numerically compared their values, the reason we did not include those treatments was that there was the chance of horizontal transmission of MG from the vaccine-injected treatments to the diluent-injected treatment via the injection hole in the shell.      

  1. What's the difference between tables 2 and 3? What is new in table 3 in relation to table 2?

Answer:

Table 2 refers to the non-injected control, and the 3 dosage treatments injected into the AC and AM. Table 3 refers to the main and interactive means comparisons of dosage and site of infection.

  1. Do you need tables 6 and 7 to show your results?

Answer:

Thank you for the comments. In those tables we compared treatments that contained vaccine or vaccine-injected with non-injected.  

  1. Table 5 – what's AN?

Thank you for the comment, AN was supposed to be AM. The relevant correction was made in Table 5.

Answer:

  1. Results titles are uninteresting to the reader. Example: "3.1.1. Comparison of site of injection (AC and AM) treatments in the diluent-control treatment group". You could use: "The site of vaccine inoculation (amnion or shell membrane) does not result in mortality or injury to the embryos". You could change all titles.

 Answer:

The authors prefer to leave the results subtitles as general descriptions, and for the specific results that were observed to be described in the section below the subtitle.

Discussion

  1. As with the results title, I found your discussion to be very "conservative". You presented more about whether your work agrees or disagrees with the results of other authors who worked with the F strain than the discussion of your results.

Answer:

This study is at the primary stage, although we found very promising results. The reason that comparisons were mainly restricted to FMG is that FMG was the only MG vaccine that has been previously administered in layers by in ovo injection, and its subsequent impacts for the incubation and growing phase were properly investigated. Therefore, we provided appropriate references to compare our findings with those for FMG. Additionally, we presented possible reasons for promising results using 6/85 MG by in ovo injection. 

  1. Strain F is a virulent strain, so we can cite this strain but not compare it in all times.

Answer:

We did not only compare our results with FMG, we also compared with ts-11 MG (Lines 431-136) and other injected vaccines or nutrients (Lines 513-520). However, the major comparisons are with FMG. This is due to the fact that FMG is virulent and 6/85 is avirulant and we needed to show that 6/85 MG can be successfully administrated with an acceptable level of protection without having negative effects on hatching quality as well as posthatch performance.   

  1. For example: Line 423, you cite Elliot and Branton and use the word "inconsistente": Your study is not inconsistent with this author. You just used an non virulent strain.

Answer:

The authors wanted to stress that the HI pipped embryonic mortality, and hatched chick mortality percentages in the current study were approximately different from those reported by Elliott et al. [29], and in that aspects were relatively inconsistent with current results. 

  1. You should focus on the advantages of your results: You can use the strain 6/85 via AM in ovo at a dosis of 173 or 1,73x10CFU/chicken embryo safely and with good antibody titers after 3 weeks of age.

Answer:

Our recommendation was the 1.73x10CFU-AM (medium dosage) treatment. However, ‎the 1.73 dosage did not show appropriate levels of antibody titers in 3 week-old pullets. Lines 442-449 and 445-455, are the examples for the effectiveness of the 1.73 x 10 102 CFU-AM treatment.  

  1. I quoted the dose of 173 because in this dose we didn't have a lesion or dead chicken embryo, your number of MG was zero at 21 days age but with a title of antibody similar to dose of 1,73x10and 1,73x104. So why did you not discuss the dose of 173 CFU?

Answer:

It is worth-mentioning that the 1.73 CFU dosage resulted in similar levels of IgG but no IgM, which indicates that this dosage would not be a good indicator for the longer period of time. In addition, it is always better to use a higher dosage of vaccine for proper immunization as long as adverse effects on performance are not observed. These are the reasons that we decided to recommend the 1.73 x 10CFU-AM and not the 1.73 CFU-AM treatment.

  1. Figure 1: I think figure 1 should be put in supplementary material.

Answer:

 It is common practice to show the clinical signs of a disease for a better understanding of the level of infection. Thus the authors prefer to include the picture in the manuscript

Conclusion

  1. Your conclusion looks like a discussion. The decision lacks objectivity.

Answer:

Thank you for the comments. However, we clearly suggested that the 1.73x10CFU-AM treatment is recommended for the vaccination and we also aimed to do further research to show how effective the in ovo injection of 6/85 MG is as compared to field vaccination, on the humoral immunity and performance of pullets.

  1. You should use up 3 lines for the conclusion. For example:

Answer:

 We agree the conclusion contains 14 lines. A paraghraph is a common practice for providing concluding statement.

  1. The Mycoplasma gallisepticumstrain 6/85 can be a secure and efficient vaccine in ovo when inoculated in fluid amnion at a dose of 1,73x10CFU/chicken embryo.

Answer:

We believe it can be safe at 1.73x10CFU, when injected into the amnion.

Reviewer 2 Report

Major comments

Abstract

·      Line 39-44 – Results description as “increase or “decrease” should be followed by the specific P values to enable the reader to determine the accurate levels of statistical significance.

Introduction

·      Line 54-55 – “path-ogenic of the avian” there is something missing here. Pathogenic what ?

Materials and methods

·      Line 126-130 – Treatment description at each level (setter, hatcher, and barn) could be presented with better clarity, perhaps presenting with the aid of a figure could help.

·      Line 146-149 – should be considered as a pre-trial to confirm site of injection, this should have been done before the main trial. Would this mean you had 29 eggs/ tray for your main trial?

·      Line 151-156 – The accuracy of the trial design and analysis need to be reconsidered. The diluent injected treatment groups (AC and AM) are reported to be placed in a separate incubator and analyzed separately, compared to vaccine injected treatment groups, for the reported reason of preventing contamination. Conversely, the non-injected eggs are placed in the same incubators as the vaccine injected treatment groups. Considering that the use of an experimental positive control treatment (in this case the diluent-injected treatment group) helps to confirm that any treatment variability observed is due to the injected treatments and not due to the “injection-vehicle i.e., diluent” or injection method. this should have been taken into cognizance. Although authors cited papers suggesting that results from non-injected control eggs and diluent-injected control should be the same, this might not be so, as experimental variabilities are unique to different studies. Going forward, authors are encouraged to consult a statistician on experimental designs that permit the comparison of all treatments together, perhaps having all treatments in multiple incubators and blocking for incubator or a split-plot design may suffice.

·      Line 212 – Compared to study cited (Sokale et al. 2017) that utilized 7 replications, this study only utilized 4 replicate cages for the grow-out trial. This needs to be mentioned as a study limitation, as it could limit the robustness and statistical power to detect significance, especially for the growth-performance parameters.  

Results

·      Table 1 -  do confirm that 1 x 10-6 is “173” or “1.73” at every other occurrence throughout the manuscript.

·      Line 280-281 – the foot notes explaining your mean separation superscripts seems confusing. This could be revised for clarity and the specific mean separation test utilized provided.  

·      Line 286-289 – This difference in mean is not specified in table 3, as I comment below.

·      Table 3 – dosage effect p value is stated to be 0.01 yet results on mean separation test are not presented. For example, in table 5, result on mean separation test was presented for “pre-pipped embryo mortality” as affected by “Injection Site” but not for other parameters in the same table. This should be corrected for all relevant tables throughout the manuscript.

·      Additionally, I believe when interaction effects are significant, only the simple effects should be presented. Conversely, when interaction effect is not significant, only the main effects should be presented. The authors appear to be misapplying means separation tests and ignoring the implications of significant main effect interactions. While a significant effect should be the trigger for means separation tests (such as Tukey’s), the significance of the interaction effect should be determining factor in whether the main effect or simple effect means are used. This applies to all result tables.

Discussion

·      Line 404-405 – The sentence “…serological response of layers” appears to be incomplete.- response of layers to what ?

Minor comments

Abstract

·      Line 23 – You could use the right multiplication symbol. There is also a space missing between 1012 and “to”

·      Line 24 – You  could also add the multiplication symbol between “1.73” and  “104”. This should apply to all instances in the manuscript.

·      Line 24 – “found that vaccinating 6/85 MG of chicks in ovo amnion at a higher dose” the phrase would need to be revised for clarity. Something along the line of “…found that the in ovo vaccination of 6/85 MG to chicks via the amnion at a higher dose…” might suffice.

·      Line 23-27 – the entire sentence starting with “The current 23 study found…” would need to be revised into smaller clear sentences. The phrase “an chick posthatch”needs needs revision.

·      Line 27- “2” in “1.73 x 102 CFU” needs to be a superscript as presented in other instances.

·      Line 25 and 28 – Abbreviations “HI” and “AM” have not been defined previously in this section.

·      Line 30 – “humor-al” should be “humoral”

Introduction

·      Line 75 – “as well as the” could be “compared to”

Materials and methods

·      Line 153 – there seems to be an extra space before “in order …”. Check lines 210, 224, 230, 263, 410, 476 for the same occurrence

·      Line 214 – close the bracket at the end of “2014”

Discussion

·      Line 428 – could be a paragraph of its own

References

·      Do confirm the author guidelines. I believe in-text multiple authors references, especially more than 3 authors should be “X et al. [ ]” and not “X, Y [ ]” as you do throughout the manuscript.

Author Response

Reviewer 2:

Major comments

Abstract

  1. Line 39-44 – Results description as “increase or “decrease” should be followed by the specific P values to enable the reader to determine the accurate levels of statistical significance.

Answer:

 The relevant corrections were applied in the Abstract.

Introduction

  1. Line 54-55 – “path-ogenic of the avian” there is something missing here. Pathogenic what ?

 Answer:

 The relevant corrections were applied.

Materials and methods

  1. Line 126-130 – Treatment description at each level (setter, hatcher, and barn) could be presented with better clarity, perhaps presenting with the aid of a figure could help.

Answer:

The authors feel that if additional figures were added, the descriptions already provided in the text would still have to be provided for clear descriptions of the experimental designs in the 3 phases. The added figures would not really add any benefit in this case. This might add more confusion.  

  1. Line 146-149 – should be considered as a pre-trial to confirm site of injection, this should have been done before the main trial. Would this mean you had 29 eggs/ tray for your main trial?

Answer:

Thank you for the suggestion. It is common practice to have some eggs injected with dye to confirm the accuracy of the machine. We have been doing this method to make sure over 90% of target sites actually received treatment. The numbers of eggs used for this test were 2 eggs from each tray that were incubated (72 overall eggs).

  1. Line 151-156 – The accuracy of the trial design and analysis need to be reconsidered. The diluent injected treatment groups (AC and AM) are reported to be placed in a separate incubator and analyzed separately, compared to vaccine injected treatment groups, for the reported reason of preventing contamination. Conversely, the non-injected eggs are placed in the same incubators as the vaccine injected treatment groups. Considering that the use of an experimental positive control treatment (in this case the diluent-injected treatment group) helps to confirm that any treatment variability observed is due to the injected treatments and not due to the “injection-vehicle i.e., diluent” or injection method. this should have been taken into cognizance. Although authors cited papers suggesting that results from non-injected control eggs and diluent-injected control should be the same, this might not be so, as experimental variabilities are unique to different studies. Going forward, authors are encouraged to consult a statistician on experimental designs that permit the comparison of all treatments together, perhaps having all treatments in multiple incubators and blocking for incubator or a split-plot design may suffice. 

Answer:

Thank you for the suggestion which is very valuable to us. The reason that we did not let diluent-injected control groups hatch with vaccine-injected treatments is that there was chance of horizontal transmission from the vaccine injected treatments to the diluent-injected treatment. For additional information, we did incubate diluent non-injected controls with vaccine-injected treatments. We incubated non-injected controls along with vaccine-injected treatments because previously we did not identify any horizontal transmission into non-injected eggs when they were incubated with FMG-injected treatments.  

  1. Line 212 – Compared to study cited (Sokale et al. 2017) that utilized 7 replications, this study only utilized 4 replicate cages for the grow-out trial. This needs to be mentioned as a study limitation, as it could limit the robustness and statistical power to detect significance, especially for the growth-performance parameters.  

 Answer:

Thank you for the suggestion and we agree with the reviewer’s comment. The relevant corrections were inserted in the text

“For the growing phase each, of 9 treatment combinations had 4 replicates which accommodated the design of the facility used for this study.” 

Results

  1. Table 1 -  do confirm that 1 x 10-6 is “173” or “1.73” at every other occurrence throughout the manuscript.

Answer:

 1 x 10-6 refers to a dilution. In order to make it more clear and consistent, the relevant corrections were applied.

  1. Line 280-281 – the foot notes explaining your mean separation superscripts seems confusing. This could be revised for clarity and the specific mean separation test utilized provided.  

Answer:

The relevant corrections were applied.

  1. Line 286-289 – This difference in mean is not specified in table 3, as I comment below.

 Answer:

The relevant corrections were performed as follows:

“However, HI was decreased, and pipped embryo and hatched chick mortalities were increased in the 1.73 x 104 CFU-AM treatment combination as compared to all other treatment combinations”.

  1. Table 3 – dosage effect p value is stated to be 0.01 yet results on mean separation test are not presented. For example, in table 5, result on mean separation test was presented for “pre-pipped embryo mortality” as affected by “Injection Site” but not for other parameters in the same table. This should be corrected for all relevant tables throughout the manuscript.

Answer:

 In Table 3, the main effect differences were significant, but the interaction differences were also significant. Because of that, no main effect results were reported for those variables. For instance in Table 4, the interaction for mortality was significant.  Table 5 refers to only diluent controls and Table 6 refers to the 9 treatment combinations where there was no significant interaction. In Table 7 there was no significant main effect, and only mortality had a significant for interaction. In Table 8, the interaction was significant for 3 wk DNA and 3 wk IgM, and there were only the main effects for Hatch DNA and 3 wk IgG.

  1. Additionally, I believe when interaction effects are significant, only the simple effects should be presented. Conversely, when interaction effect is not significant, only the main effects should be presented. The authors appear to be misapplying means separation tests and ignoring the implications of significant main effect interactions. While a significant effect should be the trigger for means separation tests (such as Tukey’s), the significance of the interaction effect should be determining factor in whether the main effect or simple effect means are used. This applies to all result tables.

Answer:

Thank you for the suggestion. There were 2 sets of data included: 1) two way ANOVA when only vaccines were compared in which we had in ovo and site of injection treatments. 2) One-way ANOVA when the non-injected treatment was compared to the vaccine treatments where we cannot compare non-injected with site of injection. Thus, we had the 9 treatment combination. To sum up, the treatment combinations including 9 treatments are different than those in the interaction effects. In addition, the authors agree with using a more conservative test such as Tukey when several treatments used. We did try our analysis with Tukey and it did not change the results. 

Discussion

  1. Line 404-405 – The sentence “…serological response of layers” appears to be incomplete.- resp onse of layers to what ?

Answer:

The serological report refers only specifically addresses the response to the 6/85 MG. The confusion for this may be due to the hyphen that connects words of the sentence given.

 “ The aim of the current study was to determine the effects various dosages of the in ovo injection of the 6/85 MG vaccine on the hatching process, early live performance, and the serological response of layers.”

Minor comments

Abstract

  1. Line 23 – You could use the right multiplication symbol. There is also a space missing between 1012and “to” 

Answer:

The relevant correction was performed.

  1. Line 24 – You  could also add the multiplication symbol between “1.73” and  “104”. This should apply to all instances in the manuscript.

Answer:

Thank you for the suggestion. A multiplication symbol is used to indicate “1.73 x 104”. We corrected all inconsistencies throughout the manuscript.

  1. Line 24 – “found that vaccinating 6/85 MG of chicks in ovo amnion at a higher dose” the phrase would need to be revised for clarity. Something along the line of “…found that the in ovo vaccination of 6/85 MG to chicks via the amnion at a higher dose…” might suffice.

Answer:

The relevant correction was performed.

  1. Line 23-27 – the entire sentence starting with “The current 23 study found…” would need to be revised into smaller clear sentences. The phrase “an chick posthatch”needs needs revision.

Answer:

 The relevant correction was performed.

  1. Line 27- “2” in “1.73 x 102 CFU” needs to be a superscript as presented in other instances.

Answer:

The relevant correction was performed.

  1. Line 25 and 28 – Abbreviations “HI” and “AM” have not been defined previously in this section.

Answer:

The relevant correction was performed.

  1. Line 30 – “humor-al” should be “humoral”

 Answer:

The relevant correction was performed.

Introduction

  1. Line 75 – “as well as the” could be “compared to”

 Answer:

The relevant correction was performed.

Materials and methods

  1. Line 153 – there seems to be an extra space before “in order …”. Check lines 210, 224, 230, 263, 410, 476 for the same occurrence

 Answer:

The relevant corrections were applied.

  1. Line 214 – close the bracket at the end of “2014”

 Answer:

 The relevant correction was performed.

Discussion

  1. Line 428 – could be a paragraph of its own

 Answer:

The relevant correction was performed.

References

  1. Do confirm the author guidelines. I believe in-text multiple authors references, especially more than 3 authors should be “X et al. [ ]” and not “X, Y [ ]” as you do throughout the manuscript.

Answer:

Thank you so much for the comment. The relevant corrections were performed.

Round 2

Reviewer 1 Report

Dear author

You have improved the material and method, but there are still numerous mistakes in your work. I advise the editor to invite a third reviewer to evaluate your work.

Abstract

The abstract improved; however, you should correct this sentence: " On d 21, 6 birds per treatment replicate were swabbed and bled."

Please put clear that "On d 21" is 21 days old of chick

Sugestion:

" At 21 days old (d), 6 birds per treatment replicate were swabbed and bled". As English is not my native language, please, if my English is not adequate, consider correcting it.

Look this sentense:

"The 1.73 × 104 43 6/85MG CFU dosage injected in the AM decreased hatchability of injected eggs containing viable embryos (HI; P=0.009) and was associated with a significant increase in late dead mortality (P=0.001). Hatchling and 3-week-old chick mortalities (P=0.008) were significantly greater in the 1.73 46 × 104 CFU-AM treatment group in comparison to all other treatment groups. In addition, 6/85MG-AM (WHICH DOSE?) treatment had no negative effects on the hatching process or on posthatch growth, and it was (???) more effective than the 1.73 x 102 6/85MG-AM treatment in the protection of pullets against MG (P≤0.0001) as compared to the low dosage and non-injected treatment groups.

Please see my comments in yellow. Were you referring to the dose of 1.73 × 104? Was this dose harmful to the embryo or not?

Introduction

About the previous correction:

"Line 107 – I didn't understand why you compared L-ascorbic acid. Was it injected via amnio?"

I understand your response, but you should explain better to the readers. I believe this section is out of context in this place in the text. However, if you can not change, maybe you can use "For example" when citing the MDV and/or ascorbic acid.

About the previous correction:

"Mycoplamas gallisepticum is not a common disease in chicks (Young birds), so why is it important to vaccinate birds in ovo? Please explain it to the reader in the introduction."

I understood your response, but please insert it in the text.

Material and method

In previous correction:

LIne 119 – the birds were vaccinated to MG? In my country, vaccinating breeding hens for MG is not allowed. However, we do not have this information.

Please insert the information about the sanitary status of the eggs (or mother flock) on the test.

In the previous correction:

Line 136 – Table 1 is not necessary.

I understand this table is trivial information that adds nothing to the article. But if the authors want to keep it, insert it in the supplementary table and not in the body of the article.

In the previous correction:

Line 141 – how many eggs were removed at 12 doi?

The information about the removed eggs was important in the previous manuscript since the number of eggs you inoculated with the vaccine was unclear. In this new version, I could understand that you used 240 viable eggs per treatment.

I think tables 2a and 2b are ok, but please, remove them from the main manuscript and insert them in the supplementary material.

In the previous review:

Line 207 (item 2.4)  -  Were there 216 birds per group?

You inoculated in 240 eggs per group. Right? And the total of born birds were 216 per group?

Results

Tables (all tables) – Please put the meaning of abbreviations at the foot of the tables.

All tables or figures should be understood without the reader returning to the text. Please insert the abbreviations as previously requested.

Table 2 – it's missing the group 173CFU-AM

In the previous review: Tables 3, 4, 7, 8 don't provide as much information when you separate dosage and place of inoculation. Therefore, I think just the information about 9 groups is necessary. If you disagree with me, please, explain why.

For me, this separation is not necessary. But if the authors insist, let them. But it doesn't bring any new information and makes your table more "polluted".

Discussion and Conclusion

I wouldn't say I liked your discussion or conclusion. I found your responses vague and weak. I spent a lot of time reviewing your article. If my corrections are not mostly accepted for the authors, I would advise the editor to look for a third reviewer to review your article.

Author Response

Reviewer 1-R3

  1. You have improved the material and method, but there are still numerous mistakes in your work. I advise the editor to invite a third reviewer to evaluate your work.

 Answer:

 The study employs 2 different experimental designs for 2 different incubators, which may not be the usual case for many studies. Because 2 different incubators were used to compared 2 different groups of treatments, this may have caused some confusion. It should be noted that in one incubator non-injected eggs were incubated with those receiving 6/85 MG treatments. In the other incubator, only the sites of injection (AC vs. AM) were compared for diluent-injected eggs. This was to prevent possible cross contamination. Nevertheless, treatment replication (8) was the same in both incubators, with the 8 replications on each of 7 treatment tray levels in one incubator and with 2 treatments on each of 8 replicate tray levels in the other incubator. However, 30 individual eggs were consistently represented in each treatment replication in both incubators.

As previously requested by the reviewer, a more specific and accurate explanation was given in the second response with an associated table to be to reflect the explanation. The information accurately reflects the experimental designs of the study. Furthermore, some highlighted additions have been included in the Materials and Methods section and in Table 3. However, the authors are unable to further provide any possible other changes to meet further complaints by the reviewer.     

Abstract

  1. The abstract improved; however, you should correct this sentence: "On d 21, 6 birds per treatment replicate were swabbed and bled."

Answer:

The above sentence was removed, because our birds from each treatment replicate group (8 per treatment) were swabbed at hatch and 3 weeks posthatch. This information was previously provided on lines 39-41.

  1. Please put clear that "On d 21" is 21 days old of chick

Sugestion:

" At 21 days old (d), 6 birds per treatment replicate were swabbed and bled". As English is not my native language, please, if my English is not adequate, consider correcting it.

Answer:

See response to comment #2.

  1. Look this sentense:

"The 1.73 × 104 43 6/85MG CFU dosage injected in the AM decreased hatchability of injected eggs containing viable embryos (HI; P=0.009) and was associated with a significant increase in late dead mortality (P=0.001). Hatchling and 3-week-old chick mortalities (P=0.008) were significantly greater in the 1.73 46 × 104 CFU-AM treatment group in comparison to all other treatment groups. In addition, 6/85MG-AM (WHICH DOSE?) treatment had no negative effects on the hatching process or on posthatch growth, and it was (???) more effective than the 1.73 x 102 6/85MG-AM treatment in the protection of pullets against MG (P≤0.0001) as compared to the low dosage and non-injected treatment groups.

Please see my comments in yellow. Were you referring to the dose of 1.73 × 104? Was this dose harmful to the embryo or not?

Answer:

The relevant changes were applied in Abstract.

Introduction

About the previous correction:

  1. "Line 107 – I didn't understand why you compared L-ascorbic acid. Was it injected via amnio?"

I understand your response, but you should explain better to the readers. I believe this section is out of context in this place in the text. However, if you cannot change, maybe you can use "For example" when citing the MDV and/or ascorbic acid.

Answer:

We aimed to compare the site of injection as well as vaccine treatments. All commercial vaccines and a majority of nutrients have been in ovo-administrated to date in the amnion. We provided other of information source to demonstrate the possibility of in ovo injection in the air cell. The only nutrient that we found to be effective when in ovo-injected into the air cell was L-AA. Thus, L-AA is a successful representative of in ovo administration in the air cell.  

About the previous correction:

  1. "Mycoplamas gallisepticumis not a common disease in chicks (Young birds), so why is it important to vaccinate birds in ovo? Please explain it to the reader in the introduction."

I understood your response, but please insert it in the text.

Answer:

The relevant changes are applied in the Introduction.

Material and method

In previous correction:

  1. LIne 119 – the birds were vaccinated to MG? In my country, vaccinating breeding hens for MG is not allowed. However, we do not have this information.

Please insert the information about the sanitary status of the eggs (or mother flock) on the test.

 Answer:

The eggs in this study were obtained from a commercial source that certified that the eggs were from an MG-clean layer flock. This information was provided on line 126 of the Materials and Methods section.

In the previous correction:

Line 136 – Table 1 is not necessary.

  1. I understand this table is trivial information that adds nothing to the article. But if the authors want to keep it, insert it in the supplementary table and not in the body of the article.

 Answer:

The authors prefer to have the table where it is presently.

In the previous correction:

  1. Line 141 – how many eggs were removed at 12 doi?

The information about the removed eggs was important in the previous manuscript since the number of eggs you inoculated with the vaccine was unclear. In this new version, I could understand that you used 240 viable eggs per treatment.

I think tables 2a and 2b are ok, but please, remove them from the main manuscript and insert them in the supplementary material.

Answer:

Because of their importance, the authors also prefer to keep those tables where they are currently.

In the previous review:

  1. Line 207 (item 2.4)  -  Were there 216 birds per group?

You inoculated in 240 eggs per group. Right? And the total of born birds were 216 per group?

Answer:

The percentage of hatched chicks in each treatment replicate group was the important aspect, and this information was provided.

Results

  1. Tables (all tables) – Please put the meaning of abbreviations at the foot of the tables.

All tables or figures should be understood without the reader returning to the text. Please insert the abbreviations as previously requested.

Answer:

All necessary abbreviations were given in all tables, and those abbreviations such as SEM is a generally accepted term by journals. This would allow us to leave it as they are.

  1. Table 2 – it's missing the group 173CFU-AM

Answer:

We do not have 173 CFU-AM treatment. This was supposed to be 1.73 CFU-AM treatment which was corrected throughout the manuscript.

  1. In the previous review: Tables 3, 4, 7, 8 don't provide as much information when you separate dosage and place of inoculation. Therefore, I think just the information about 9 groups is necessary. If you disagree with me, please, explain why.

Answer:

Please see response to comment # 1 for a specific explanation of the experimental design.

  1. For me, this separation is not necessary. But if the authors insist, let them. But it doesn't bring any new information and makes your table more "polluted". 

Answer:

The separations were added according to what was requested in the previous review.

Discussion and Conclusion

  1. I wouldn't say I liked your discussion or conclusion. I found your responses vague and weak. I spent a lot of time reviewing your article. If my corrections are not mostly accepted for the authors, I would advise the editor to look for a third reviewer to review your article.

Answer:

Thank you for your review. We have done all that we know possible to address the concerns.

Reviewer 2 Report

Line 46- Close the bracket after the P value 

Specific P values should also be applied to the results section. 

Line 132-147- Contrarily, the description of the methods in text and the use of a figure/table are not mutually exclusive but rather complementary. Below are references to in ovo studies that have used a table or figure to perfectly describe their methods. I still believe this would better explain the treatment allocation and description in this study. 

Subramaniyan, S.A., Kang, D.R., Park, J.R., Siddiqui, S.H., Ravichandiran, P., Yoo, D.J., Na, C.S. and Shim, K.S., 2019. Effect of in ovo injection of l-arginine in different chicken embryonic development stages on post-hatchability, immune response, and Myo-D and myogenin proteins. Animals9(6), p.357.

Nouri, S., Ghalehkandi, J.G., Hassanpour, S. and Aghdam-Shahryar, H., 2018. Effect of in ovo feeding of folic acid on subsequent growth performance and blood constituents levels in broilers. International Journal of Peptide Research and Therapeutics24, pp.463-470.

Line 146-149- This is why I consider the use of a chart to explain the experimental treatment/layout expedient. You do mention that 72 eggs were used for the experiment. I couldn't come around how the numbers came up to 72. I suppose you had 9 treatments, 8 replicate trays, 30 eggs/ tray ?

Line 149-160- According to your response "we did incubate diluent non-injected controls with vaccine-injected treatments." What group is the "diluent non-injected controls" ?

Tables- Result presentation. Perhaps authors did not fully understand my comment,  I am in no way suggesting the type of mean separation test to be carried out. I am only commenting on the presentation of such. Your main effects are sites (AC, AM) and dosage (1.73 CFU, 1.73 x 102 CFU, 1.73 x 104 CFU). Your simple effect are each dosage response under AC and AM respectively, and your interaction compares everything together. Hence my comment was "when interaction effects are significant, only the simple effects should be presented. Conversely, when interaction effect is not significant, only the main effects should be presented."  Perhaps the following links below can help provide broader perspective on my comment. 

https://www.youtube.com/watch?v=z6_6AmsO4y4

https://vault.hanover.edu/~altermattw/courses/220/spss/2way/2way-2a.html

Additionally (relevant to all result tables), If authors do decide to present results as it is, you cannot deliberately choose which means to apply a mean separation test to and which to ignore, as you do for HI (Table 3, P= 0.01, no mean separation), Hatched chick morality (Table 4, P=0.02, no mean separation). 

Author Response

  1. Line 46- Close the bracket after the P value 

Answer:

The relevant correction was applied.

  1. Specific P values should also be applied to the results section. 

Answer:

Thank you for the suggestion, however, the exact P-values have been shown in the relevant tables and also it is clearly indicated in the Materials and Methods section that the cut off for significant P-values is 0.05. Also, we provided the exact P-values that were close to the significant level. For example, on line 314 when the P-value was 0.058, on line 337 when the P-value was 0.057, on line 382 when the P-value was 0.074, and on line 413 when the P-value was 0.068. It is common practice to not report the P-values in the Results section. 

  1. Line 132-147- Contrarily, the description of the methods in text and the use of a figure/table are not mutually exclusive but rather complementary. Below are references to in ovo studies that have used a table or figure to perfectly describe their methods. I still believe this would better explain the treatment allocation and description in this study. 

Subramaniyan, S.A., Kang, D.R., Park, J.R., Siddiqui, S.H., Ravichandiran, P., Yoo, D.J., Na, C.S. and Shim, K.S., 2019. Effect of in ovo injection of l-arginine in different chicken embryonic development stages on post-hatchability, immune response, and Myo-D and myogenin proteins. Animals9(6), p.357.

Nouri, S., Ghalehkandi, J.G., Hassanpour, S. and Aghdam-Shahryar, H., 2018. Effect of in ovo feeding of folic acid on subsequent growth performance and blood constituents levels in broilers. International Journal of Peptide Research and Therapeutics24, pp.463-470.

Answer:

The second paragraph in section 2.2. of the Materials and Methods was rewritten to include revised and added information to more specifically and accurately describe the experimental visual design (see highlighted lines). Also, Tables 2.a. and 2.b. were added to give a corresponding description of the treatment arrangements in hatchery units.

  1. Line 146-149- This is why I consider the use of a chart to explain the experimental treatment/layout expedient. You do mention that 72 eggs were used for the experiment. I couldn't come around how the numbers came up to 72. I suppose you had 9 treatments, 8 replicate trays, 30 eggs/ tray?

 Answer:

Thank you for the correction. A separate group of 64 total eggs (1 egg from each injection treatment replicate group) were injected with dye to confirm the site of injection [8 injection treatment (non-injected eggs not included) x 8 replicates]. “72” was changed to “64” to indicate that only injected eggs were included.

  1. Line 149-160- According to your response "we did incubate diluent non-injected controls with vaccine-injected treatments." What group is the "diluent non-injected controls" ?

 Answer:

We apologize for the confusion. There were no "diluent non-injected controls". We meant “non-injected controls”, in which non-injected controls were incubated with vaccine-injected treatments. Diluent-injected controls were incubated in a separate incubator.  

  1. Tables- Result presentation. Perhaps authors did not fully understand my comment,  I am in no way suggesting the type of mean separation test to be carried out. I am only commenting on the presentation of such. Your main effects are sites (AC, AM) and dosage (1.73 CFU, 1.73 x 102 CFU, 1.73 x 104 CFU). Your simple effect are each dosage response under AC and AM respectively, and your interaction compares everything together. Hence my comment was "when interaction effects are significant, only the simple effects should be presented. Conversely, when interaction effect is not significant, only the main effects should be presented."  Perhaps the following links below can help provide broader perspective on my comment. 

https://www.youtube.com/watch?v=z6_6AmsO4y4

https://vault.hanover.edu/~altermattw/courses/220/spss/2way/2way-2a.html

Answer:

In addition to interaction mean separations, main effect mean separations were also added to Tables 4, 5, 8, and 9. Although, as suggested, when there was a significant interaction, only those interaction sub-class means differences were discussed. However, when there was no significant interactions, but when there were main effect differences, only the main effect mean differences were discussed.

  1. Additionally (relevant to all result tables), If authors do decide to present results as it is, you cannot deliberately choose which means to apply a mean separation test to and which to ignore, as you do for HI (Table 3, P= 0.01, no mean separation), Hatched chick morality (Table 4, P=0.02, no mean separation). 

Answer:

Please see the response to comment #6.

Round 3

Reviewer 1 Report

As I wrote in the last correction, the authors did not revise almost anything I asked for in the discussion the first time around. And, again, they ask not to change several of my notes.

Definitely, this article cannot be published as it is. It is an interesting article but the quality of the results, discussion, and conclusion is not good.

Reviewer 2 Report

Just a minor observation:

Line 132-133: "eggs injected in the AC or AM with Poulvac Marek’s disease diluent (Zoetis, Parsippany, NJ) alone as controls". What is the dose of the injected vaccine ?

Author Response

Reviewer #2-R2

  1. Line 132-133: "eggs injected in the AC or AM with Poulvac Marek’s disease diluent (Zoetis, Parsippany, NJ) alone as controls". What is the dose of the injected vaccine?

Answer:

The eggs were not injected with any particular type of vaccine, but were injected with Marek’s disease (MD) diluent. Currently, the MD diluent has been used for many injected materials including nutrients and several vaccines. In addition, MD diluent is the only solution used to suspend vaccines for commercial in ovo injections. Furthermore, to date, there has been no other specific solution produced for the suspension of the 6/85 MG vaccine.   

Round 4

Reviewer 2 Report

Thanks for the clarification.